# Identification of *TPX2* Gene Family in Upland Cotton and its Functional Analysis in Cotton Fiber Development

**DOI:** 10.3390/genes10070508

**Published:** 2019-07-04

**Authors:** Kang Lei, Aiying Liu, Senmiao Fan, Huo Peng, Xianyan Zou, Zhang Zhen, Jinyong Huang, Liqiang Fan, Zhibin Zhang, Xiaoying Deng, Qun Ge, Wankui Gong, Junwen Li, Juwu Gong, Yuzhen Shi, Xiao Jiang, Shuya Zhang, Tingting Jia, Lipeng Zhang, Youlu Yuan, Haihong Shang

**Affiliations:** 1State Key Laboratory of Cotton Biology, Key Laboratory of Biological and Genetic Breeding of Cotton, The Ministry of Agriculture, Institute of Cotton Research, Chinese Academy of Agricultural Sciences, Anyang 455000, Henan China; 2Zhengzhou Research Base, State Key Laboratory of Cotton Biology, Zhengzhou University, Zhengzhou 450000, China

**Keywords:** upland cotton, Xklp2 (*TPX2*) gene family, gene expression, MAP, protein interactions

## Abstract

Microtubules (MTs) are of importance to fiber development. The Xklp2 (TPX2) proteins as a class of microtubule-associated proteins (MAPs) play a key role in plant growth and development by regulating the dynamic changes of microtubules (MTs). However, the mechanism underlying this is unknown. The interactions between TPX2 proteins and tubulin protein, which are the main structural components, have not been studied in fiber development of upland cotton. Therefore, a genome-wide analysis of the *TPX2* family was firstly performed in *Gossypium hirsutum* L. This study identified 41 *GhTPX2* sequences in the assembled *G. hirsutum* genome by a series of bioinformatic methods. Generally, this gene family is phylogenetically grouped into six subfamilies, and 41 *G. hirsutum*
*TPX2* genes (GhTPX2s) are distributed across 21 chromosomes. A heatmap of the *TPX2* gene family showed that homologous *GhTPX2* genes, *GhWDLA2/7* and *GhWDLA4/9*, have large differences in expression levels between two upland cotton recombinant inbred lines (69307 and 69362) that are different in fiber quality at 15 and 20 days post anthesis. The relative data indicate that these four genes are down-regulated under oryzalin, which causes microtubule depolymerization, as determined via qRT-PCR. A subcellular localization experiment suggested that *GhWDLA2* and *GhWDLA7* are localized to the microtubule cytoskeleton, and *GhWDLA4* and *GhWDLA9* are only localized to the nucleus. However, only *GhWDLA7* between *GhWDLA2* and *GhWDLA7* interacted with *GhTUA2* in the yeast two-hybrid assay. These results lay the foundation for further function study of the *TPX2* gene family.

## 1. Introduction

Cotton is a principal source of natural fibers, and cotton fiber development can be divided into four overlapping stages: Initiation, elongation, secondary wall deposition, and maturation [1]. Fiber cells which are single-celled trichomes without cell division, provide an excellent biological model for studying cell elongation and synthesis of secondary cell walls [2]. The microtubule cytoskeleton plays an essential role in fiber development [3]. The arrangement of the cortical microtubules is associated with elongation and synthesis of secondary cell walls of fiber cells [4]. Microtubules (MTs) are mainly formed by heterodimers of α-tubulin and β-tubulin concatenated end-to-end [5,6]. γ-tubulin is not a major structural component of microtubules, but can mediate microtubule nucleation at microtubule organizing centers [7]. Tubulins are involved in the regulation of microtubule assembly and function, and a reduction of α-tubulin can disrupt the microtubule structure and further cause abnormal cell expansion [8]. Six α-tubulin genes (*TUA*), nine β-tubulin genes (*TUB*), and two γ-tubulin genes have been discovered in Arabidopsis [9,10,11]. A study of overexpression of *GhTUA9* in fission yeast [3] showed that *GhTUA9* was involved in fiber elongation.

Microtubule-associated proteins (MAPs), which bind to MTs, are involved in MT functions by regulating microtubule dynamics, organization, and the establishment of polarity, wall construction, signal transduction and so on [9,12,13,14,15]. The orientation and dynamic changes in MTs are controlled by the interaction between MTs and MAPs. These proteins, such as MAP18, MAP20, MAP65, and MAP70, suggest that MAPs mediate dynamic changes in microtubule arrays, the material transportation, and the interaction between MTs and other structures in plant growth and development [14,16,17,18,19,20,21]. MAP18 regulates cortical microtubule organization by destabilizing associated microtubules to modulate polar cell growth in vegetative tissues [22]. These critical biological roles indicate that MAPs are worthy of studying in plants. IQ67 DOMAIN5 (IQD5), as a class of plant-specific MAPs, regulates MT dynamics that affect MT organization and subsequent cell shape formation in Arabidopsis [23].

The Xklp2 (TPX2) proteins that belong to the MAPs participate in the formation and development of MTs [24]. TPX2 proteins bind Aurora kinases to regulate spindle formation by the N-terminal Aurora binding domains, and bind microtubules affect cell division by the C terminus [24]. TPX2 proteins which contain a highly conserved domain (Pfam:PF06886) are conserved in higher plants. Some TPX2 proteins have been reported, such as TPX2, MAP20, and WDL in plants [25]. The over-expression of the *EgMAP20* gene in Arabidopsis results in organ twisting. *EgMAP20* and *EgWDL3* were localized to the microtubule cytoskeleton [26]. During the period of transition of elongation and secondary wall synthesis, the rate of fiber elongation decreases rapidly [27,28]. During this period, the MTs shift from transverse to oblique and helical orientations, indicating that MTs play a key role in this process [29,30]. When the growth rate of fiber cell elongation decreases, the orientation of MTs change [31]. However, the cotton *TPX2* genes and the relationship between TPX2 proteins and tubulin are less known, and this gene family is less understood in cotton (*Gossypium hirsutum* L). In our study, the authors analyzed phylogeny relationships, gene structures, chromosomal locations, expression patterns, and preliminary functional analysis. Further, the authors preliminarily illustrated the relationship between GhTPX2 proteins and GhTubulins. The information about the *GhTPX2* gene family may be useful for further studies.

## 2. Materials and Methods

### 2.1. Plant Material and Abiotic Stress Treatments

Two upland cotton genotypes 69307 with longer and stronger cotton fiber and 69362 with short and weaker cotton fiber were cultivated under standard field conditions (10 rows, each 8 m long and 0.8 m) in Zhengzhou, China [32,33]. Cotton fibers at 5, 10, 15, 20, 25, and 30 days post anthesis (DPA) were collected from bolls for extraction of RNA. Cotton bolls were harvested at 0 DPA for ovule culture in vitro, surface sterilized with 75% (*v/v*) ethanol for 10 min, and immersed into 95% (*v/v*) ethanol for 5 s, then flamed briefly using an alcohol lamp. The ovules were cultured on a BT medium [34]. Ovules were transferred to the medium containing 10 µM oryzalin for 8 days after 10 days of cultivation. This study used 0.1% (*v/v*) dimethyl sulfoxide (DMSO) as the solvent control. MTs in cultured ovule fibers were observed by confocal microscopy, and cotton fibers were collected from ovules for RNA extraction. *Nicotiana benthamiana* plants were grown in the greenhouse with a 16 h light/8 h dark cycle at 22 °C, and 60% relative humidity light intensity of 100 120 μmol·m^−2^ · S^−1^.

### 2.2. Identification and Sequence Retrieval of TPX2 Genes

This study identified the plant *TPX2* (PF06886) gene family members using the hidden Markov model profile of the HMMER3.0 program [35] using the published genomes of the following seven species: *Arabidopsis thaliana* [36], *Theobroma cacao* [37], *Populus trichocarpa* [38], *Gossypium hirsutum* [39], and *Gossypium raimondii* [40], *Gossypium barbadense* [39], and *Gossypium arboretum* [41]. The Blastp program with *AtTPX2* sequences as the query was used (*e*-value of 10^−5^), and SMART (http://smart.embl-heidelberg.de/) database were used to remove the redundant sequences. These *GhTPX2* genes were named following a similar nomenclature approach [42].

### 2.3. Phylogenetic Tree Construction

The plant TPX2 sequences containing seven plant species, *A. thaliana*, *T. cacao*, *P. trichocarpa*, *G. hirsutum*, *G. raimondii*, *G. barbadense*, and *G. arboreum*, were aligned using the ClustalW program. A neighbor-joining (NJ) phylogenetic tree was constructed using the MEGA6.06 program [43] with 1000 bootstrap replicates.

### 2.4. Gene Expression Analysis

The RNA-seq data of seven different tissues (root, stem, leaf, flower, ovule, seed, and fiber) from *G. hirsutum* TM-1 was downloaded from NCBI Gene Expression repository (https://www.ncbi.nlm.nih.gov/bioproject/PRJNA248163/). The RNA-sequencing data for various stages of fiber development from two upland cotton lines 69307 and 69362 (10, 15, 20, 25, 30 DPA) were downloaded from the NCBI Gene Expression repository (http://www.ncbi.nlm.nih.gov/bioproject/ PRJNA508480) [32]. The SRA data was converted to a fastq format using the SRA Toolkit with the –split-3 parameter. The reference genome sequences for *G. hirsutum* were downloaded (https://www.cottongen.org/) [39], and were used to construct libraries with the Bowtie2 program [44]. The fastq data were inspected using the FastQC program and mapped to the *G. hirsutum* genomes with the default parameters of the TopHat2 program [45]. The FPKM values of *TPX2* genes were calculated using the Cufflinks program [46], with BAM files sourced from TopHat2 and the parameters, –library-type and fr-unstranded (Appendix A). The gene expression data (FPKM) were divided by the mean of all the values, then were normalized with the log2 (FPKM+1) method to calculate the expression levels. Gene expression patterns between different fiber developments were visualized with heat maps using the R (3.3.0) software package (https://CRAN.R-project.org/package=pheatmap).

### 2.5. Gene Structure Conserved Motifs Analysis and Chromosomal Localization

This study obtained the structural information of the GFF3 files of *TPX2* genes. Exons and introns of *G. hirsutum TPX2* family genes were predicted using Gene Structure Display Server2.0 (http://gsds.cbi.pku.edu.cn/) [47]. The online program MEME (http://meme-suite.org/) was used to identify the conserved motifs in *TPX2* genes with the following parameters: 15 motifs and a motif width of 6–50 amino acids [48]. The chromosomal distribution of the *TPX2* genes was confirmed based on genome annotations. Mapchart 2.2 software was used to visualize the distribution of the *TPX2* genes on the *G. hirsutum* chromosomes [49].

### 2.6. RNA Isolation and the qRT-PCR Analysis

The total RNA was extracted from the cotton fiber using the RNA simple Total RNA Kit (BioTeke, Beijing, China). The quality of RNA was detected using a NanoDrop 2000 spectrophotometer. All samples were stored in liquid nitrogen and maintained at –80 °C. Then, using the RNA as the template, every sample contained 1 μg RNA. Following this, the cDNA was reverse transcribed using a First Strand cDNA Synthesis Kit (Takara Biotechnology Co, Ltd., Dalian, China). The gene-specific primers used for qRT-PCR were designed with a primer database (http://biodb.swu.edu.cn/qprimerdb). (Appendix A). The authors used the *GhHistone3* gene as an internal control, and a total volume of 20 μL that contained 0.4 μL of each primer (10 μM), 2 μL cDNA, 10 μL SYBR Premix Ex Taq (2×), and ddH_2_O to make up the volume used to perform qRT-PCR with three biological and technological replicates on a LightCycler480 system (Roche) using SYBR Premix Ex Taq (Takara). The reaction parameters were as follows: 95 °C for 5 min, 40 cycles of 95 °C for 10 s, 60 °C for 10 s, and 72 °C for 10 s, followed by a melt carve from 60 to 95 °C, which was executed in a 96-well plate. The gene relative expression levels were calculated using the 2^−△△Ct^ method [50].

### 2.7. Subcellular Localization of TPX2 Proteins

The coding sequence (CDS) of *GhWDLA2*, *GhWDLA4*, *GhWDLA7*, and *GhWDLA9* were amplified. Then recombinant plasmids (GhWDLA4-GFP, GhWDLA9–GFP, GhWDLA7-GFP, and GhWDLA2-GFP) were generated using In-fusion cloning technology, and transformed into *Agrobacterium* (GV3101). A confocal microscope (OLYMP-USFV1200) with excitation at 514 nm, and scanning at 520–555 nm was used to observe three-week-old tobacco leaf epidermal cells, 48 to 72 h after infiltration [51].

### 2.8. Gene Duplication and Selection Pressure

This study used an all-versus-all BLASTp search (*e*-value < 10^−5^) to detect orthologous and paralogous gene pairs using the MCScan program [52,53]. The circos program [54] was used to visualize the circular maps of the gene pairs. The genes that were separated by five or fewer genes in 100-kb chromosome fragment were regarded as tandem duplicated genes [55]. Non-synonymous Substitution Rate/Synonymous Substitution Rate (Ka/Ks) of selected homologous genes were calculated using the PAL2NAL web server (http://www.bork.embl.de/pal2nal#RunP2N).

### 2.9. Yeast Two-Hybrid (Y2H) Assay

The Y2H system [56] was used to detect the interactions between *TPX2* proteins and tubulin proteins. Generally, the open reading frame (ORFs) of *GhWDLA4*, *GhWDLA9*, *GhWDLA7*, and *GhWDLA2* were constructed into a prey vector, pGADT7. The coding sequence of *TUA2* and *TUA9* which were preferentially expressed in fiber, were constructed into a bait vector, pGBKT7. The constructed vectors in the appropriate combinations were transformed into yeast cells. The transformed yeast cells were transferred to the quadruple dropout medium: SD/–Ade/–His/–Leu/–Trp supplemented with X-a-Gal and Aureobasidin A (QDO) to detect the pair-wise interactions.

## 3. Results

### 3.1. Identification of TPX2 Genes in the G. hirsutum Genome

A total of 170 TPX2 protein sequences from seven species were identified (Additional File 1) after removing redundancy sequences. This study identified 41 genes whose protein sequences have TPX2 (PF06886) domains from *G. hirsutum*. Among the 41 *GhTPX2* genes, 20 genes were from the At genome and 21 genes were from the Dt genome. Further, 41 genes were identified from *G*. *barbadense*, 21 genes from *G. aboreum,* and 21 genes from *G. raimondii*. The number of tetraploid *TPX2* cotton genes is approximately twice as many as diploid cotton genes, which suggests that there is no *TPX2* cotton gene loss after ployploidization. The coding lengths of *G. hirsutum*, *G. aboreum,* and *G. raimondii* of *TPX2* genes ranged from 450 to 2466 bp. The encoded protein sequences of TPX2 proteins ranged from 149 to 821 amino acids. Except for GhWDLC1, GaWDLB2, and GrWDLC5, the predicted isoelectric point (pI) of all TPX2 proteins were more than seven. This indicates that cotton TPX2 proteins are rich in basic amino acids (Appendix A).

### 3.2. Bioinformatic Analysis of Plant TPX2 Family Proteins

In total, 170 *TPX2* genes were confirmed in the genomes of *G. hirsutum*, *G. barbadense*, *G. arboreum*, *G. raimondii*, *T. cacao*, *P. trichocarpa*, and *A. thaliana*. A multiple alignment analysis was performed using the neighbor-joining method. A phylogenetic tree was constructed to study the evolutionary relationships (Figure 1, Additional file 2). The TPX2 protein sequences were clustered into six main groups which were similar to previous studies [26]. According to a previous study, Clades 1–3 were separately named MAP20 (Clade1), MAP20L (Clade2), and TPX2 (Clade3) [57,58,59]. As clades 4–6 mainly contain WDL proteins, the number of cotton family members is higher than *AtPTX2*. Therefore, the authors named these clades WDLA (Clade4), WDLB (Clade5), and WDLC (Clade6) to distinguish the different subfamilies. All these sequences have the TPX2 domain, which can interact with microtubules, and influence microtubule dynamics or assist with different microtubule functions [24]. The WDLB subgroup (Clade 4) has the largest number of members. The MAP20 subgroup (Clade 1) has the fewest members. Notably, among these seven species, every species contains six subfamily numbers (Clade1–6).

### 3.3. Gene Structure Conserved Motifs Analysis 

This study analyzed the exons and introns arrangement to further investigate the phylogenetic relationships among different members of the *TPX2* gene family. The *TPX2* family has different gene lengths in the subfamilies. The gene length of *GhMAP20_1* is the shortest (1700 bp), and the gene length of *GhWDLC6* is the longest (6700 bp). There was a considerable difference in the exon-intron structure between different groups. Most of the genes had six to nine exons and some genes had 19 exons (Figure 2). The MEME program was used to identify 20 conserved motifs (Figure 2). Motif 1 and 2 compose the TPX2 domain. Every GhTPX2 member has motif 1 and 2. Motif 12 has a copper fist DNA binding domain. All subfamily (clade 4) members except GhWDLA2 and GhWDLA7 contain motif12. Some *GhTPX2* genes have different motif structures, however, most homologous *GhTPX2* genes have the same motif structure, such as *TPX2_1/TPX2_3*, *GhWDLA2/GhWDLA7*, *GhWDLA4/GhWDLA9*, and *GhMAP20_1/GhMAP20_2.* Most members from the same subfamily have similar motif features, exon-intron structure, and gene length, supporting the close evolutionary relationships.

### 3.4. Chromosomal Location, Gene Duplication, and Selection Pressure

Among 41 *GhTPX2* genes, 20 and 21 genes are localized to the 11 chromosomes from At subgenome and 12 chromosomes from Dt subgenome, respectively. Almost, all of *TPX2* are located on the proximate or the distal ends of the chromosomes (Figure 3). Among the 21 chromosomes, most chromosomes have one or two *TPX2* genes. However, chromosomes A03 and D02 have four genes. It can be inferred that genetic variation existed in the progress of evolution, evidenced by this unbalanced distribution of the *GhTPX2* genes on the chromosomes. During the gene family evolution, tandem duplication and segmental duplication contributed to the generation of the gene family to some extent [60]. Our data showed that all members of *TPX2* genes expanded only by segmental duplication (Appendix A). This result suggests that segmental duplication played a key role in the evolutionary progress of the *TPX2* gene family. Circular maps were used to visualize the syntenic relationships among *G. arboretum*, *G. raimondii*, and *G. hirsutum* (Figure 4). The Ka:Ks ratio were used to assess whether homologous genes were under positive selection pressure (Ka:Ks > 1) or purifying selection pressure (Ka:Ks < 1). Therefore, PAL2NAL was used to calculate the Ka:Ks ratio of the *TPX2* gene family. The ratios for *TPX2* genes were lower than one, which indicates that purified selection was crucial to these homologous gene pairs (Appendix A).

### 3.5. Expression Analysis of TPX2 Family Genes in Cotton

To understand the specific functions of *TPX2* genes in fiber development, this study performed the expression patterns analyses. Firstly, the RNA-sequencing data that were downloaded were used to detect the expression profiles of 41 *TPX2* genes in different tissues (root, stem, leaf, flower, ovule, and fiber) using a heatmap that was generated using the R (3.3.0) software package. As shown in Figure 5a, *GhWDLA4*, *GhWDLA6*, *GhWDLA9*, *GhWDLA7*, *GhWDLA2*, *GhWDLB1*, *GhWDLB9,* and *GhWDLB4* had higher expression levels in fibers relative to the other tissues. Secondly, a transcriptome analysis of the *TPX2* genes was performed using the RNA-seq data for different stages of fiber development for *G. hirsutum* (69307 and 69362). The expression patterns of genes provide distinct clues to functional divergence. A heatmap was used to display the *TPX2* gene expression patterns during different fiber developments for 69307 and 69362 (Figure 5b). The analysis revealed that most of the genes from the WDLA and WDLB subfamilies had high expression levels. Among them, only *GhWDLA4*, *GhWDLA9*, *GhWDLA7*, and *GhWDLA2* had obvious differences at different stages of fiber development, and they were all highly expressed at 20 DPA, which was a critical time for fiber strength in 69307 compared to 69362. The expression levels of the four genes at 20 DPA were approximately double at 15 DPA and 25 DPA in 69307. However, this phenomenon was not observed in 69362. The data suggest that these genes have a high expression level at 20 DPA in 69307. This study also analyzed the expression level of tubulin genes, which were described previously [3,61]. A heat map showed that *GhTUA2*, *4*, *9*, and *10* had a high expression level in 69307 at 20 DPA compared to 69362. This implies that these genes may also be related to secondary cell wall biosynthesis (Figure 5c).

The expression levels of *GhWDLA4*, *GhWDLA9*, *GhWDLA7*, and *GhWDLA2* in different fiber developments for 69307 and 69362 were detected using qRT-PCR. As shown in Figure 6, the relative expression data showed that the expression levels of *GhWDLA2*, *GhWDLA7*, *GhWDLA4,* and *GhWALA9* were the highest at 20 DPA in 69307 and 69362.

### 3.6. Expression Profiles of TPX2 Genes under Abiotic Stresses

As shown in Figure 8a, most fiber microtubules grown in BT medium containing oryzalin were destroyed. Oryzalin is a kind of microtubule-disrupting agents that cause microtubule depolymerization [62]. The expression of the four genes after microtubule depolymerization was further tested. The relative expression data suggest that treatment with oryzalin results in a different degree of reduction in the expression levels of the four genes (Figure 7b).

### 3.7. Subcellular Localization of TPX2 Proteins

Considering the structure and expression patterns of these genes, this study performed the subcellular localization of selected TPX2 proteins in *G. hirsutum*. The green fluorescent protein (GFP)-tagged *TPX2* proteins were temporarily expressed in tobacco leaf epidermal cells, and visualized by a confocal laser-scanning microscope. As shown in Figure 6a,b, GFP-GhWDLA4, and GFP-GhWDLA9 were only located in the nucleus. The net-like structures throughout the leaf epidermal cell indicate that GFP-GhWDLA2 and, GFP-GhWDLA7 are exclusively distributed along the MT cytoskeleton (Figure 8c,d).

### 3.8. Interactions Between TPX2 Proteins and Tubulins

As α-tubulin and β-tubulin are the principle components of microtubules, and Xklp2 (TPX2) proteins are a kind of microtubule-associated protein. These two genes that are specifically localized to the microtubule cytoskeleton were examined to determine if they could interact with tubulins (*GhTUA2, 9,*) which were found to be preferentially expressed in fibers at 20 DPA for 69307 using a pair-wise interaction in the Y2H assay. As shown in Figure 9, yeast cells containing pGADT7::*GhWDLA7* and pGBKT7::*GhTUA2* produced blue coloration on the medium: SD/–Ade/–His/–Leu/–Trp supplemented with X-a-Gal and Aureobasidin A. This indicates that this combination interacts.

## 4. Discussion

The Xklp2 (TPX2) proteins that belong to the microtubule-associated family of proteins can interact with microtubules to regulate microtubule dynamics or assist with different microtubule functions [26]. Microtubule-associated proteins (MAPs) specifically bind to the MT cytoskeleton to regulate their dynamic changes and be involved in MT functions [63]. As a class of MAPs, Xklp (TPX2) proteins considerably affect plant growth and development [39]. Although the whole-genome sequences of upland cotton were determined, a genome-wide analysis of *TPX2* gene family had not been performed in cotton [39]. Our study identified 41 *GhTPX2* genes in the AD1 genomes, 41 *GbTPX2* genes in AD2 genomes, 21 *GaTPX2* genes in A genomes, and 21*GrTPX2* genes in the D genomes. These results suggest that the loss of *TPX2* genes did not occur in allotetraploid *G. hirsutum*, which is not consistent with the higher rate of gene loss in allotetraploids [64,65]. This result may indicate that *TPX2* genes are conserved. According to the *AtTPX2* genes, 170 genes from seven different species are divided into six subfamilies which is similar to the previous classification in Arabidopsis [26]. Every species contains six subfamily genes for further identifying the conservation. The similar arrangements of exons, introns, and motifs in the same family further prove the correctness of the *TPX2* classifications. Most genes have six exons but some genes have more than six exons, which suggest that *TPX2* genes may have different regulatory mechanisms (Figure 2).

The expression profiling in six different tissues show that *GhWDLA4*, *GhWDLA6*, *GhWDLA9*, *GhWDLA7*, *GhWDLA2*, *GhWDLB1, GhWDLB9*, and *GhWDLB4* have higher expression levels in fibers relative to the other tissues, which suggests that these genes are preferentially expressed in fiber and may potentially function in fiber development (Figure 4a). The expression levels in different fiber developments suggest that *GhWDLA4*, *GhWDLA9*, *GhWDLA7*, and *GhWDLA2* are highly expressed at 20 DPA, especially in 69307. Fibers rapidly elongate during 5–20 DPA and during the late stage of fiber elongation. The onset of secondary cell wall biosynthesis occurs approximately from 16–21 DPA, depending on the cotton species and the environment [66]. Further, 18–21 DPA is key for cell wall remodeling and synthesis of the winding layer [67,68]. Therefore, these genes were regarded as highly expressed genes in 69307 at 20 DPA, an important time for fiber secondary wall synthesis. Consequently, it is assumed that these genes may be involved cell secondary wall synthesis to influence fiber strength by the interaction with MTs (Figure 4b).

According to qRT-PCR, *GhWDLA2* and *GhWDLA7* expression reached the highest level at 20 DPA in 69307. *GhWDLA4* and *GhWALA9* had low expression levels, but still reached the highest level at 20 DPA in 69307. This is generally consistent with RAN-seq data. However, the expression levels of *GhWDLA4* and *GhWALA9* were not completely consistent with RNA-seq data. The difference may be induced by various factors. These four genes were all from of *GhWDLA* (clade 4), which suggest that this subfamily may play a critical role in fiber development. The relative expression levels of these four genes are down-regulated after microtubule depolymerization, which might further suggest an interaction between these genes and microtubules (Figure 8). TPX2 has been shown to localize to the microtubules in the interphase, and it may decorate other microtubule arrays during the other stages of plant cell division [26]. Therefore, this study investigated the subcellular localization of these four genes in epidermal cells of tobacco leaf, and found that homologous *GhWDLA4/9* was expressed in the nucleus and homologous *GhWDLA2/7* was specifically localized to the MT cytoskeleton. Thus, the authors assume that *GhWDLA2,* and *GhWDLA7* genes may be involved in fiber secondary cell wall synthesis to influence fiber strength by the interaction with MTs. Compared with 69362, *GhTUA2*, *4*, *9*, and *10* are highly expressed at 20 DPA in 69307, which is an important time related to fiber strength with longer and stronger cotton fiber, which indicates that they may be related to fiber strength. The expression patterns of these genes are similar to *GhWDLA2, 4, 7,* and *9.* Additionally, the related data indicated that *GhTUA2* was expressed until 20 DPA [69]. *GhTUA2/9* forms a distinct branch on the phylogenetic tree and *GhTUA9* may affect fiber elongation [3]. Therefore, the Y2H assay with different combinations between *GhWDLA2/7* and *GhTUA2/9* was performed. The Y2H assay showed that *GhWDLA7* interacted with *GhTUA2*, but not with *GhTUA9*. Our results suggest that *GhWDLA7* may be involved in the regulation of fiber strength through interaction with *GhTUA2*. Although, *GhWDLA2* and, *GhWDLA7* are localized to microtubules, only *GhWDLA7* interacted with *GhTUA2* in the Y2H assay. This difference may be caused by other reasons. This result indicates that although these proteins are relatively conserved, the potential regulatory mechanisms of these genes are different.

## 5. Conclusions

This study performed a genome-wide analysis of the phylogeny of the *TPX2* gene family in upland cotton. The expression patterns in different fiber development stages for two upland cotton lines (69307 and 69362) that have different fiber quality indicated that *GhWDLA2, GhWDLA7, GhWDLA4,* and *GhWDLA9* might be involved in fiber cell wall synthesis. Relatively, expression levels after microtubule depolymerization suggest that these genes are closely related to microtubules. However, only *GhWDLA2,* and *GhWDLA7* were found to be localized to microtubules through a subcellular localization experiment. Our Y2H assay and previous studies indicate that *GhWDLA7* might play a functional role in cell wall synthesis by interacting with *GhTUA2* to influence microtubules and may ultimately affect fiber strength. These findings facilitate further investigations in the fiber development of *G. hirsutum*.

## Figures and Tables

**Figure 1 genes-10-00508-f001:**
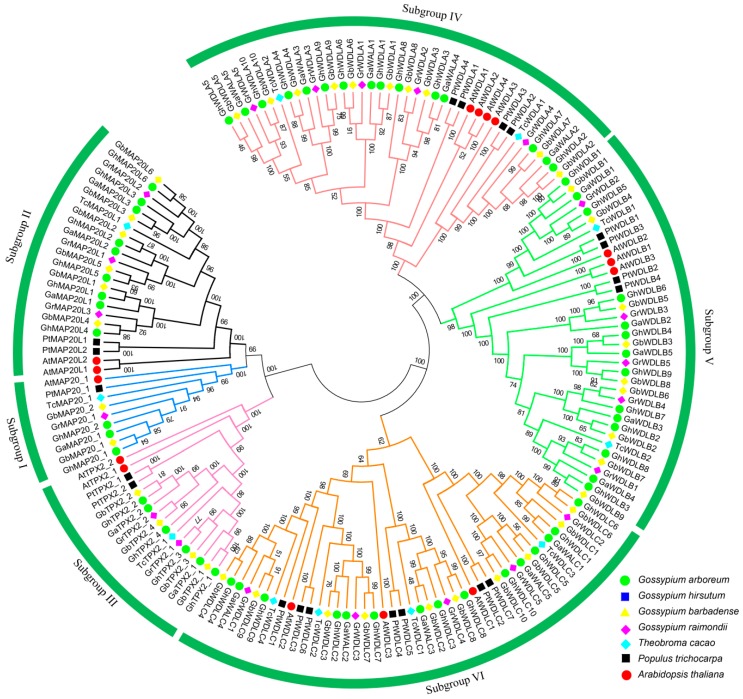
Neighbor-joining phylogenetic tree of TPX2 proteins from different plant species *Gossypium hirsutum*, *Gossypium arboreum*, *Gossypium barbadense*, *Gossypium raimondii*, *Arabidopsis thaliana*, *Populus trichocarpa*, and *Theobroma cacao*.

**Figure 2 genes-10-00508-f002:**
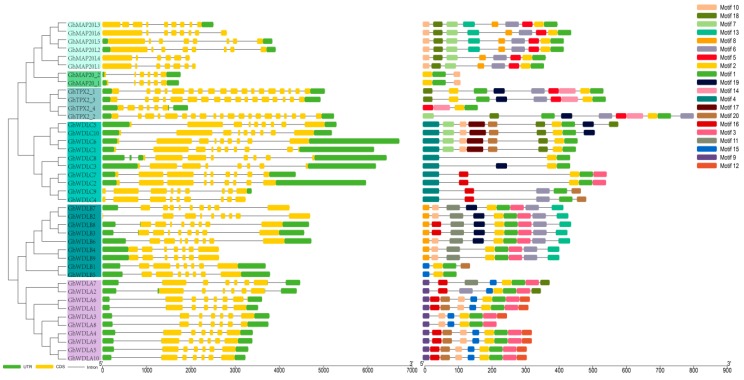
Phylogenetic relationship, gene structure predictions, and conversed protein motif analysis of *GhTPX2* family: (**a**) A neighbor-joining phylogenetic tree created using the MEGA6 program: (**b**) Gene structure analysis of *GhTPX2* genes, and (**c**) conversed protein motifs.

**Figure 3 genes-10-00508-f003:**
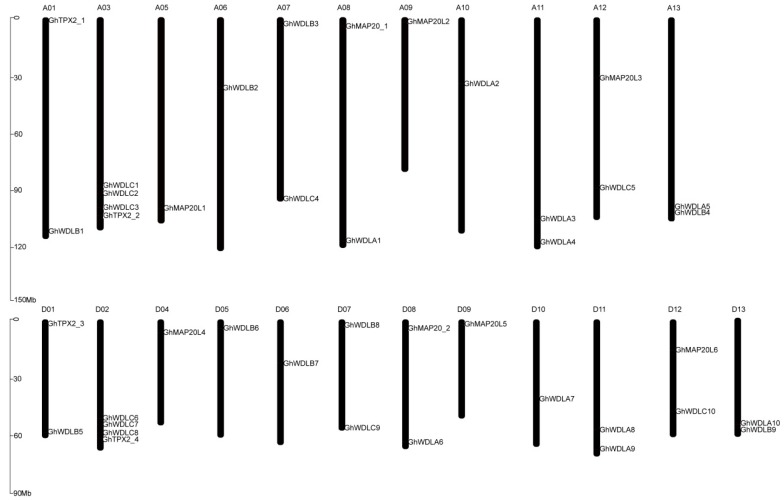
Chromosomal distribution of *TPX2* genes in *G. hirsutum* L.

**Figure 4 genes-10-00508-f004:**
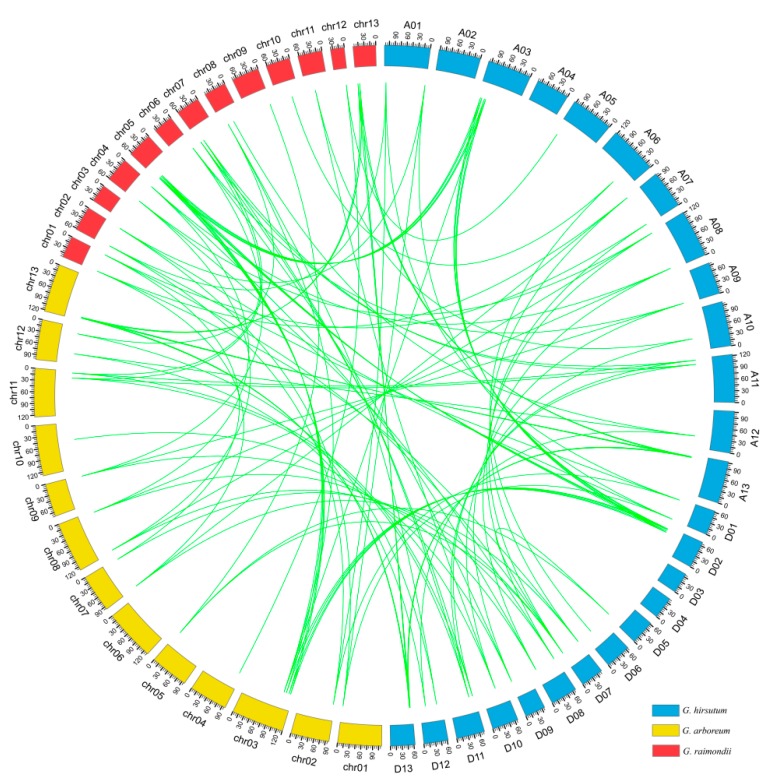
*TPX2* homologous gene pairs among *G. arboreum, G. raimondii,* and *G. hirsutum.*

**Figure 5 genes-10-00508-f005:**
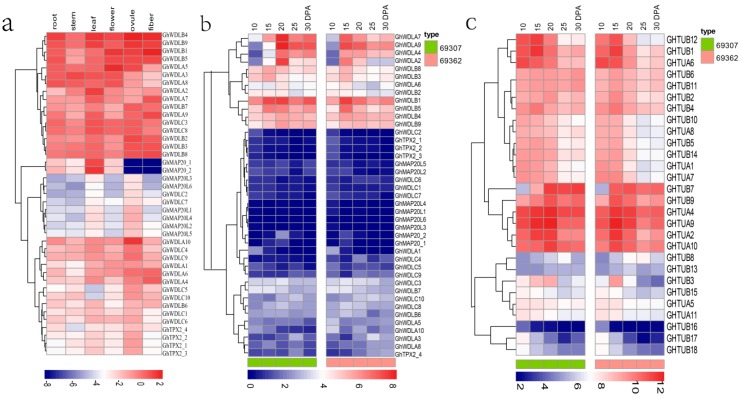
Expression analysis of *TPX2* genes in *G. hirsutum*: (**a**) heatmap showing the expression levels of *GhTPX2* genes in different tissues: (**b**) heatmap presenting the expression levels of *GhTPX2* genes, and (**c**) heatmap presenting the expression levels of *Ghtubulin* genes.

**Figure 6 genes-10-00508-f006:**
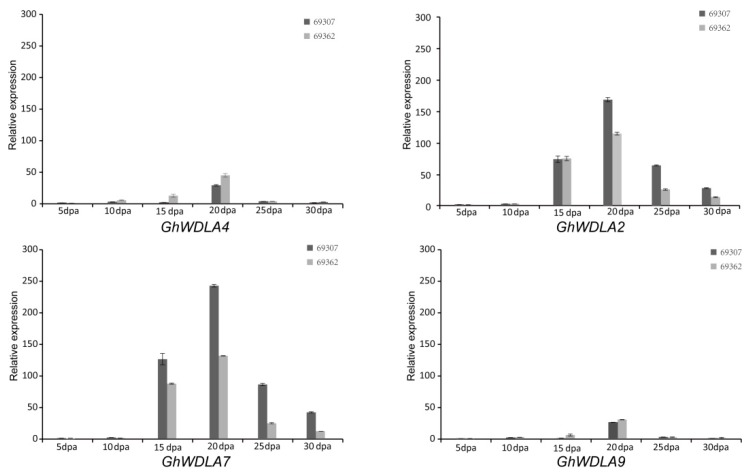
The relative expression levels of selected genes in different fiber developments in 69307 and 69362 determined using qRT-PCR. The relative expression levels were normalized against the reference gene *GhHistone3*. The error bars represent the standard deviations of three experiments.

**Figure 7 genes-10-00508-f007:**
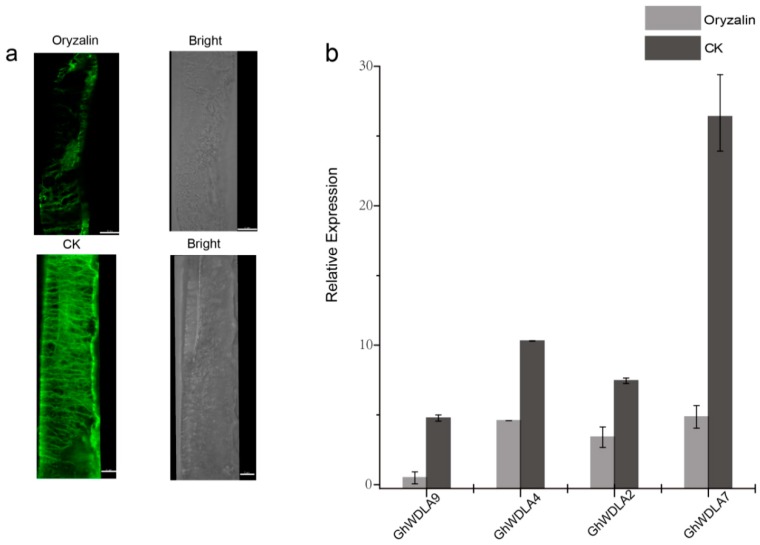
(**a**) The microtubule arrays of cotton fibers cultured on BT medium containing 10 µM oryzalin: (**b**) Expression patterns of *GhTPX2* genes (*GhWDLA2, 4, 7,* and *9* in the cotton fibers at 18 days post anthesis (DPA) after microtubules (MTs) under oryzalin). The relative expression levels are shown against the reference gene *GhHistone3*. The error bars represent the standard deviations of three experiments.

**Figure 8 genes-10-00508-f008:**
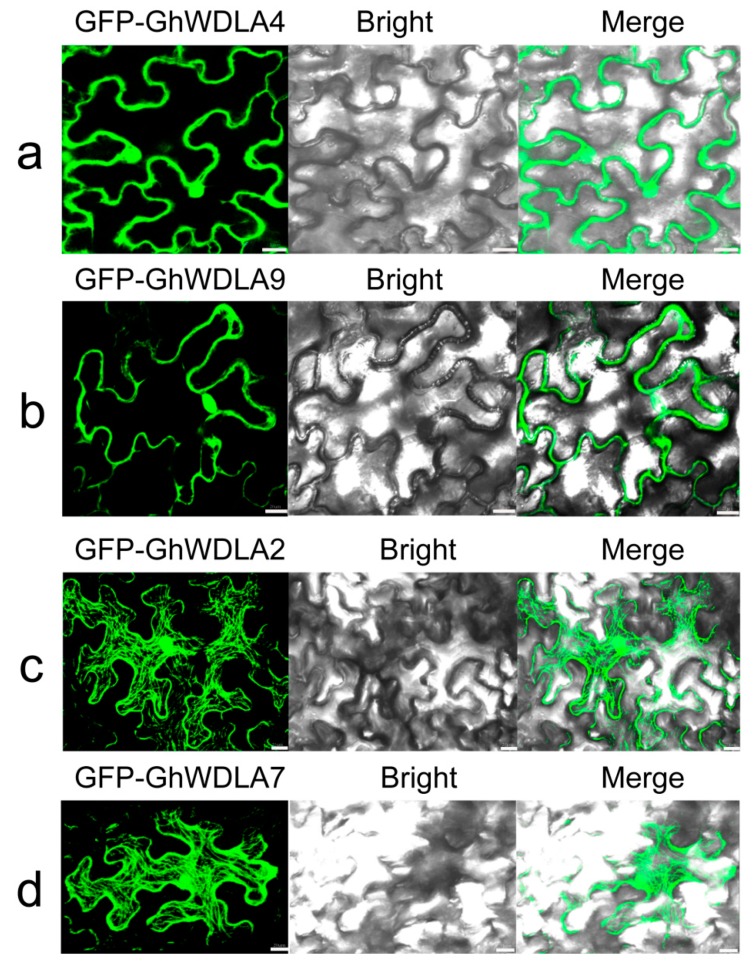
Subcellular localizations of four selected TPX2 proteins in the epidermal cell of tobacco leaf: (**a**) GFP-GhWDLA4: (**b**) GFP-GhWDLA9: (**c**) GFP-GhWDLA2, and; (**d**) GFP-GhWDLA7. Scale bar = 20 μm.

**Figure 9 genes-10-00508-f009:**
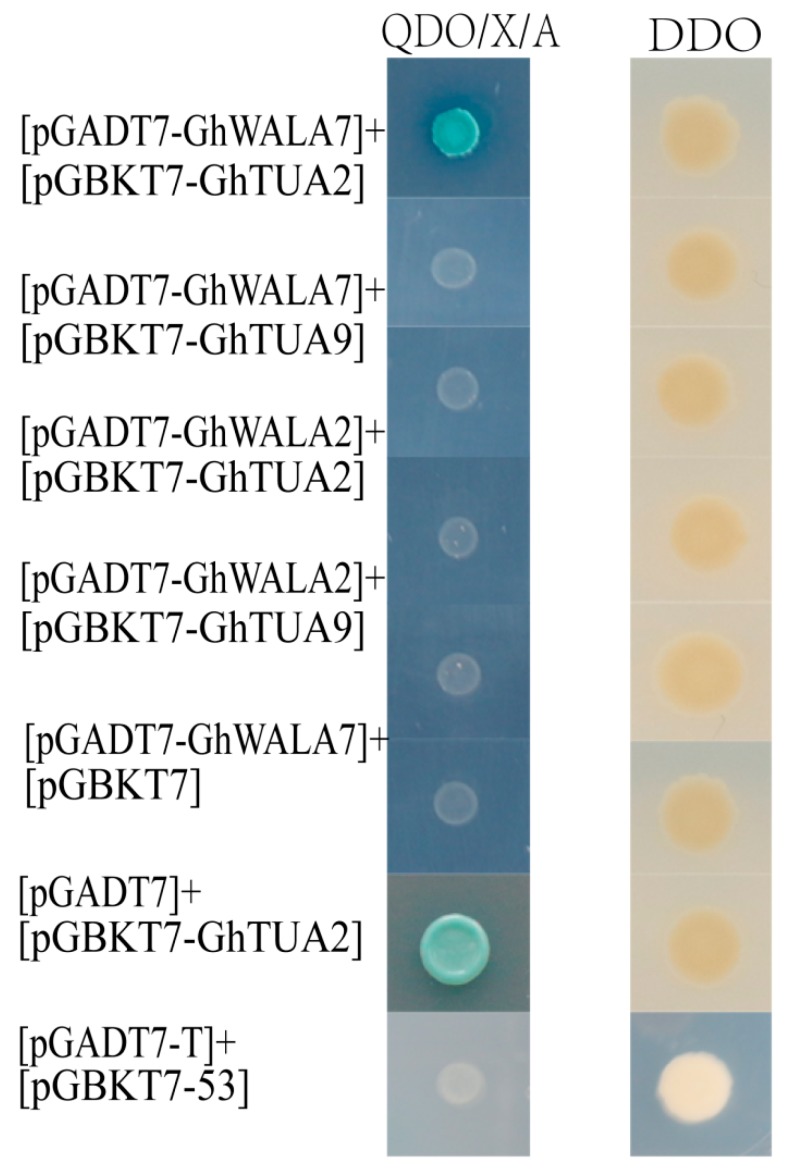
Yeast two-hybrid assay indicates that *GhWDLA7* interacts with GhTUA2: [pGBKT7-53]+[pGADT7-T] as positive. [pGBKT7-Lam]+[pGADT7-T] as negative.

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
