# Peer review of "Identification of TPX2 Gene Family in Upland Cotton and its Functional Analysis in Cotton Fiber Development"

_genes, 2019, doi:10.3390/genes10070508_

Round 1
Reviewer 1 Report
Authors of this study described TPX2 gene family in Upland cotton. They found 4 genes in this family which were differentially expressed in short- or long-fiber varieties; two of these genes were localized in microtubule cytoskeleton, whereas other two - in nucleus. The revisions suggested below may improve the quality of the manuscript.
Manuscript will benefit from professional English correction services. There are some sentences that difficult to understand. Some terms should be exchanged for better suited. For example, line 77 in methods section - “cotton strains”. Term “strain” is mainly used in bacteriology; for cotton would be more appropriate to use “variety” or “cultivar”. Another sticking example in the same section on line 86 is “conservatory”; I am guessing authors mean greenhouse or growth chamber.
Authors use a newer version of G. hirsutum genome (Wang et al., 2018), but older version of G. barbadense genome (Liu et al., 2015). Is any specific reason for this? A newer version of G. barbadense genome is available from the same group (Wang et al., 2018).
Methods. How fibers were separated from ovules at earlier developing stages?
Methods, lines 137-141: Please provide details of constructions of recombinant plasmids, such as whether the full length coding sequence or only signal peptide were used. Also please provide scanning parameters to obtain images.
Figure 6. The purpose of this figure is to compare the expression of the gene in two cotton cultivars, with short and long fiber. Therefore, it would be better to present expression of one gene in two cultivars on the same graph. You can make 4 graphs for simplicity.
Reviewer 2 Report
The results of the manuscript seem to be interesting however, there were some key problems in the manuscript. I recommend doing extensive revision and re-evaluation of the manuscript.
Please find comments in the attached revised file.

Reviewer 3 Report
The manuscript performed the analysis for TPX2 gene family in upland cotton using bioinformatic and experimental strategies. Authors stated results of TPX2 gene family from across-species perspective, then focus on this gene family in cotton, then used experimental methods to explore expressions, subcellular localization and protein interactions. However, when read through the manuscript, it is lack of basis of scientific writing. Although authors displayed a systematic analysis for their study, which may be interesting, a number of language issues may cause the manuscript to be very hard to read. I insist that authors should spend much more time on organizing language before it can be published and it will be nice to have English editors to polish the manuscript entirely. Authors are confused about using some words like “of”, “with”, …; authors should spend more time on figuring out how to organize words as a sentence; authors should pay more attention to make their text to be concise and readable; authors should know what tense should be used in different places. Such kinds of points need dramatical improvements, otherwise I don’t think readers can understand their paper. I would like to give several places as examples: Line 25-27, it is a confused sentence, should be modified; Linc 31, “However, only…”, in this sentence, “between GhWDLA2 and GhWDLA7 can interact with GhTUA2 ” is obviously superfluous; Line 50, “and so on” is a kind of clumsy word in here… It is pretty easy to catch these types of things, please be careful to go through the manuscript and correct those errors. I recommend authors to check other published papers online and use the common rule of scientific writing in their manuscript. Now I am going to start from Materials and Methods session,
1. Line 77, should use “genotypes” other than “strains”, you are studying plants but not bacteria. Where are they come from? Authors mention two types of genotypes, but not four, should explain this clearly. What is standard condition? should at least mention when you planted?
2. Line 79, authors defined DPA in Line 80 but used it in Line 79, be careful to check this similar issues in the manuscript
3. Line 86, I never saw anyone use “conservatory” in a scientific paper, if you use greenhouse, just mention it is greenhouse.
4. Line 101, just say RNA-seq data.
5. Line 133, not repeats, should use “replicates”
6. Line 146-147, definitions of Ka and Ks are totally wrong, please check and correct
7. Line 162-163, XX genes from YY, this is not a sentence.
8. Line 163-165, from reading this sentence, it seems authors make the conclusion about “diploid cotton genes which suggest that there is no gene loss after polyploidization”, it seems authors concluded this sentence for all tetraploid genes, but they just used the results from 170 TPX2? I don’t think this is a strong evidence to make the conclusion.
9. Results 3.3, authors labeled motifs numerically, but should mention more about what these motifs are. I don’t think people will understand biological meaning of “motif 1” et al.
10. Line 248, where is the figure legend for subpanel c? Heatmap but not heat map; should label the unit for the scale bar in every heatmap.
11. Figure 6, label x and y axis.
12. Figure 7, have two “Bright” subpanels, it seems one is refer to oryzalin and another one in refer to CK. But authors should clearly point out what the difference between these two. Please label y axis in subpanel b.
13. Figure 8, authors mention bar=20um, there are very dim bar scale in figures, please highlight them.
Round 2
Reviewer 3 Report
Authors have addressed most of my concerns in the manuscript properly and English writing was also improved. The manuscript looks to have a much better shape. I have several minor suggestions for authors,
1. For Ka/Ks, please defining it either way as below,
You can define Ka/Ks as Non-synonymous/Synoymous Substitution Rate
Or you can define Ka/Ks as Non-synonymous Substitution Rate/Synonymous Substitution Rate. However, Ka does not just mean non-synonymous and Ks does not just mean synonymous. Please correct this place.
2. For several places like “by using”, just say “using”, there is no need to add “by”
3. Please increasing the font size in figure 3.
